# Two Important Anticancer Mechanisms of Natural and Synthetic Chalcones

**DOI:** 10.3390/ijms231911595

**Published:** 2022-09-30

**Authors:** Teodora Constantinescu, Alin Grig Mihis

**Affiliations:** 1Department of Chemistry, Faculty of Pharmacy, Iuliu Hatieganu University, 400012 Cluj-Napoca, Romania; 2Advanced Materials and Applied Technologies Laboratory, Institute of Research-Development-Innovation in Applied Natural Sciences, “Babes-Bolyai” University, Fantanele Str. 30, 400294 Cluj-Napoca, Romania

**Keywords:** chalcones, molecular hybridization, ABCG2 cassette, tubulin, inhibitory activity, new biological molecules

## Abstract

ATP-binding cassette subfamily G and tubulin pharmacological mechanisms decrease the effectiveness of anticancer drugs by modulating drug absorption and by creating tubulin assembly through polymerization. A series of natural and synthetic chalcones have been reported to have very good anticancer activity, with a half-maximal inhibitory concentration lower than 1 µM. By modulation, it is observed in case of the first mechanism that methoxy substituents on the aromatic cycle of acetophenone residue and substitution of phenyl nucleus by a heterocycle and by methoxy or hydroxyl groups have a positive impact. To inhibit tubulin, compounds bind to colchicine binding site. Presence of methoxy groups, amino groups or heterocyclic substituents increase activity.

## 1. Introduction

Cancer is characterized by uncontrolled, progressive, rapid and pathological cell proliferation and is one of the most aggressive diseases in the world [1,2]. This condition is a correlation of pathologies characterized by uncontrolled and abnormal cell growth, which are based on gene mutations [3,4]. The altered activity of some transcription factors determines the appearance of many types of cancers [5]. Heterogeneity of cancer cells targets is associated eith genomic instability, gene variation in tumor formation and ability of tumor cells to have excessive phenotypic variations [6]. This has important clinical consequences, as tumor formations with complex subclonal structures can be aggressive and have the ability to initiate resistance to therapy and form metastases [7,8]. The formation of metastases to distant organs is often incurable and is a determining factor in the death of cancer patients, which in these cases, is frequently associated with dysfunctions of vital organs [9,10]. Cancer cells have aberrant redox homeostasis, reactive oxygen (ROS) species by being pro-tumorigenic and, at high levels of ROS, have cytotoxic activity [11,12]. Due to aberrant redox homeostasis in tumor cells and their microenvironment, these reactive species are insoluble in carcinogenesis, but also in anticancer therapy. Numerous studies have shown that the role of ROS in malignant cells, positive or negative, depends on factors such as tumor type, cancer stage, therapeutic strategies, duration of cell exposure to ROS, specificity and ROS level [13,14]. The increased number and clinical importance of molecular biomarkers in usual practice allow anticancer therapies to have an increased specificity on a particular genetic complex in tumor formation. The disadvantages of these biomarkers are related to the expected duration of analyses and the need to take samples from tissues [15]. There are four major types of approaches available for cancer treatment (radiotherapy, surgery, immunotherapy and drug therapy) [16,17,18,19,20]. Less than a quarter of patients with common cancers receive appropriate therapy [21]. Early diagnosis of cancer provides the opportunity to choose the appropriate therapeutic intervention. Among methods used for early diagnosis are tumor markers and imaging techniques, such as computed tomography, magnetic resonance imaging, position emission tomography and ultrasound scanning endoscopy (cytogenetic and cell genetics screening) and so forth [22]. Standard treatment for most cancers includes chemotherapeutic agents. The response to this type of therapy varies for each patient, especially in case of poorly characterized cancers. Tumor heterogeneity, the presence of cancer stem cells and the plasticity of these cells indicate the need to use combination therapies and/or targeted therapies [23]. Platinum compounds are a major component of chemotherapy for various types of cancer. Three platinum compounds used in anticancer therapy-cisplatin, carboplatin and oxaliplatin-have the ability to distort the normal functioning of DNA (deoxyribonucleic acid) by generating monoadducts [24]. Unlike chemotherapy, targeted therapy is considered more effective and safer for the treatment of cancer [25]. In recent years, immunotherapy has become a major alternative to conventional anticancer therapy, which aims to increase the quality of life of patients and prolong life [26]. Immunotherapy, ideal for fighting cancer, acts on the immune system to eradicate malignant cells [27]. There are two therapeutic directions of immunotherapy: blocking immune checkpoints by antibodies that block immune system inhibitory receptors in tumors, and adoptive cell therapy by which T cells are designed to express chimeric antigenic receptors [28,29,30,31]. It is also known that most recently approved targeted anticancer therapies are specific for coding oncoproteins for mutant somatic genes. These agents include proteins essential for the maintenance of specific cell descendants or proteins that inhibit immune responses, such as protein 1 of programmed cell death (PD-1) [32]. Immunotherapy, based on immunostimulatory monoclonal antibodies, explains the involvement of an immune response in anticancer therapy. PD1/PDL1 blockade demonstrates the importance of targeted immune mechanisms in tumor formation, by which T cell response in tumor formation is rehabilitated [33]. The most common problem with antitumor therapy is resistance to therapy [34,35,36,37,38,39,40,41]. Some cancer cells initiate resistance to many chemotherapeutic agents, others acquire this characteristic through mutations in the process of carcinogenesis [42]. This is correlated with cassette-binding ATP (ABC) transporters, which are P-glycoprotein (P-gP/ABCB1), multidrug resistance protein 1 (MRP1/ABCC1) and breast cancer resistance protein (BCRP/ABCG2). One method of treatment is the elimination of these resistant cancer cells, using multidrug ABC transporters [43,44,45,46,47,48,49,50,51].

Diet is known to play an important role in the prevention, onset and progression of cancer [52]. Due to remarkable chemical diversity of natural products, many biologically active compounds have been isolated from plants, microbes and other living organisms, which have been shown to have anticancer activity [53].

Numerous studies on cell cultures and animal models show that different natural compounds with various chemical structures have the ability to act as chemopreventive and chemotherapeutic agents. These biologically active molecules come from structural classes such as polyphenols, alkaloids, terpenoids and organosulfur compounds and so forth [17].

## 2. Flavonoids

Flavonoids are the largest class of low-molecular-weight polyphenolic secondary metabolites with a benzopyron residue (C6-C3-C6) in the molecule. They consist of two aromatic rings joined by a bridge of three carbon atoms that form an oxygenated heterocycle [54,55,56,57,58,59]. Over 10,000 flavonoids have been described, many of which are present in large quantities in fruits and vegetables [60]. Flavonoids are biosynthesized by phenylpropanoid, starting from phenylalanine [61]. Chalcone synthetase is the first enzyme in flavonoid biosynthesis and catalyzes the formation of chalcones from one molecule of p-coumaroyl-CoA and three molecules of malonyl-CoA [62]. Depending on the degree of oxidation of the central nucleus, flavonoids are classified into flavones, flavanones, flavonols (proanthocyanidins), dihydroflavonols, isoflavones, isoflavanones, chalcones, aurones, anthocyanidins and tannins (Appendix A) [63,64,65,66].

Flavonoid compounds are associated with a number of biological activities, their antioxidant, anticancer, anti-inflammatory, antidiabetic, antimicrobial, antidyslipidemic, neuroprotective, antiosteoporotic, cytoprotective, hepatoprotective, vasoprotective properties being much studied [67,68,69,70,71,72,73,74,75]. Flavonoids are known to decrease cardiovascular risk and cancer-associated mortality [76]. Experimental studies also show that dietary flavonoids interact with rhodopsin and modulate the functions of visual pigments [77]. The structural diversity of flavonoids is associated with chemical changes such as hydroxylation, methylation, acylation, glycosylation, etc. In plants, glycosylation is the most common modification of flavonoids, being responsible for increasing solubility and stability, maintaining bioactivity, regulating transport and accumulating and/or decreasing toxicity of these compounds [78]. By methylation of hydroxy groups of flavonoids, compounds with superior stability, a longer metabolism time, a favorable transport rate and, implicitly, a better absorption in target tissues are obtained [79]. For example, hesperidin (a natural flavanone, Appendix A, Compound **1**) has low water solubility and has low absorption in the small intestine. By methylation of hesperidin in an alkaline medium, hesperidin methyl chalcone (Appendix A, Compound **2**) is formed, a compound with superior water solubility and favorable intestinal absorption, bioavailability and tissue distribution [80]. In general, metallic complex combinations of flavonoids have superior cytotoxic, anti-inflammatory and redox properties [81].

Flavonoid compounds are associated with a number of biological activities, their antioxidant, anticancer, anti-inflammatory, antidiabetic, antimicrobial, antidyslipidemic, neuroprotective, antiosteoporotic, cytoprotective, hepatoprotective, vasoprotective properties being much studied [67,68,69,70,71,72,73,74,75]. Flavonoids are known to decrease cardiovascular risk and cancer-associated mortality [76]. Experimental studies also show that dietary flavonoids interact with rhodopsin and modulate the functions of visual pigments [77]. The structural diversity of flavonoids is associated with chemical changes such as hydroxylation, methylation, acylation, glycosylation, etc. In plants, glycosylation is the most common modification of flavonoids, being responsible for increasing solubility and stability, maintaining bioactivity, regulating transport and accumulating and/or decreasing toxicity of these compounds [78]. By methylation of hydroxy groups of flavonoids, compounds with superior stability, a longer metabolism time, a favorable transport rate and, implicitly, a better absorption in target tissues are obtained [79]. For example, hesperidin (a natural flavanone, Appendix A, Compound **1**) has low water solubility and has low absorption in the small intestine. By methylation of hesperidin in an alkaline medium, hesperidin methyl chalcone (Appendix A, Compound **2**) is formed, a compound with superior water solubility and favorable intestinal absorption, bioavailability and tissue distribution [80]. In general, metallic complex combinations of flavonoids have superior cytotoxic, anti-inflammatory and redox properties [81].

### 2.1. Chalcones

Compounds with an enone system in molecules continue to be of particular interest due to their simple chemistry, the ease with which they are obtained by chemical synthesis, their biological importance and the possibility of being precursors in organic chemistry [82]. Chalcones (chalconoids, 1,3-diaril-2-propen-1-ones) are natural and synthetic compounds in which aromatic residues are joined by an α, β-unsaturated electrophiles chain of three carbon atoms (Appendix A) [66,83,84,85,86,87,88]. Reactive hydrogens from the basic structure of chalcones determine the possibility of these compounds being structurally modified in order to obtain a large number of derivatives [89]. Chalcones are precursors for many heterocyclic compounds such as cyanopyridines, 2-pyrazolines, pyrazoles, isoxazoles, pyrimidine-2-ones, flavanones and flavones and so forth [90,91]. Data from literature indicate that molecules with a 1,3-diaryl-2-propen-1-one residue have numerous biological activities and, for this reason, are of particular pharmacological importance [92,93]. Among the most important biological activities of chalcones are antitumor, antibacterial, anti-inflammatory, antifungal, cytotoxic, antiviral, antimalarial, antituberculous, antiparasitic, antiulcer, analgesic, antipyretic, antihepatotoxic, etc., properties [94,95,96,97,98,99,100,101,102,103,104,105,106,107,108,109,110,111,112,113,114,115,116,117,118,119,120]. Chalcones have an extended conjugate system, which facilitates their binding to bioactive molecules, such as enzymes and DNA [121]. For example, chalcones are active on some enzymes such as acetylcholinesterase and carbonic anhydrase, being active in glaucoma, obesity, osteoporosis, epilepsy, Parkinson’s and Alzheimer’s [122]. The presence of α, β-unsaturated carbonyl group, which acts as a Michael acceptor, facilitates the interaction of chalcones with sulfhydryl groups in cysteine residues or with thiol groups. This interaction is considered to be responsible for the numerous biological properties of these compounds [123,124]. The biological activity of chalcones is also strongly influenced by their chemical structure, especially by substituents on two aromatic residues. It has advantages related to its ability to regulate molecular pathways related to cancer, favorably influence apoptosis, metastases and the response to cellular stress. Numerous methoxychalcones with antitumor properties have been identified, the presence of the methoxy group being favorable for this activity [125,126,127,128]. Since chalcones are active in vitro and in vivo in cancers susceptible to therapy, as well as in case of resistant ones, these compounds represent an important skeleton for the identification of new anticancer agents. In addition, 1,3-diaryl-2-propen-1-ones represent an important class of natural small molecules with anticancer chemotherapeutic action [129]. For example, chalcones are important pharmacophores for many natural products, such as flavokawaina, millepachine and xanthohumol (XN) (Appendix A, Compounds **3**–**7**) [130]. XN (Appendix A, Compound **7**), a prenylated chalcone from female hop inflorescences, inhibits the growth of MCF-7 (human breast adenocarcinoma cancer cells) and induces their apoptosis. Prenylchalcone is also effective in hepatocellular carcinoma and prostate cancer [131]. Mechanisms of the anticancer activity of XN include inhibition of initiation and progression of carcinogenesis, induction of apoptosis, and inhibition of angiogenesis [132,133]. Natural compounds with 1,3-diaryl-2-propen-1-one residue in molecules can be readily converted to various by-products by exposure to UV light. Natural chalcones can also be metabolized rapidly by drug metabolism enzymes, which causes a short half-life in vivo. For these reasons, numerous series’ of stable chalcone-like compounds have been obtained whose biological properties have been evaluated [134]. The most widely used method of obtaining chalcones is Claisen-Schmidt condensation (Appendix A) [66]. During the reaction, the base (catalyst) attacks α-hydrogen of acetophenone and forms an enolate anion (carbanion). The carbanion attacks the carbonyl group of benzaldehyde and forms a β-hydroxycarbonyl intermediate, from which chalcone is obtained [135,136,137,138,139,140,141].

Synthetic chalcones can also be obtained by Suzuki reaction, Friedel-Crafts acylation, Wittig reaction and photo-Fries rearrangement of phenyl cinnamate [142]. Among synthetic chalcones, hybrid molecules with indole in the molecule are widely used in anticancer therapy. For example, Yan J et al. synthesized a series of 29 indole chalcones in order to evaluate their antiproliferative activity. The most active compound in the series, (E)-3-(6-Methoxy-1H-indol-3-yl)-2-methyl-1-(3,4,5-trimethoxyphenyl)propen-2-en-1-one (Appendix A, Compound **8**), has IC50 values = 3–9 nM on six cancer cell lines. The new tubulin polymerization inhibitor binds to the colchicine binding site. Studies of cellular mechanisms show that derivative blocks the cell cycle in the G2/M phase and induces apoptosis consecutively with decreasing mitochondrial membrane potential. The indole compound has low cytotoxicity compared to normal human cells and has a similar potency to therapy-resistant cells [143].

Given the need to identify new anticancer therapies and the growing interest in identifying the biological properties of chalcones, we evaluated the antitumor mechanisms of these compounds.

#### 2.1.1. Anticancer Activity of Chalcones

The anticancer potential of chalcones is correlated with their ability to act on various molecular targets such as ABCG2 (ATP binding cassette subfamily G), tubulin, activated nuclear B cell growth (NF-κB), vascular endothelial growth factor (VEGF), tyrosine kinase receptor (EGFR), mesenchymal-epithelial transition factor (MET), 5-α reductase, ACP-reductase, histone deacetylase, p53, CDC25B (protein tyrosine phosphatase), retinoic acid receptors, estrogenic topoisomerase receptors and MDM2 [66,144,145,146,147]. In order to obtain chalcones with superior anticancer properties, three methods of modulation of natural chalcones were used: (1) modulation of substituents at the level of the two aromatic residues (aldehyde and acetophenone), (2) replacement of aromatic residues with heterocycles and (3) obtaining hybrid molecules by conjugating chalcones with other molecules with antitumor properties [66]. Since studies performed to highlight the anticancer activity of chalcones are numerous, and mechanisms by which they exert this property are multiple, we wanted to conduct a study on two important antitumor mechanisms of these compounds.

#### 2.1.2. ABCG2

ABCG2 (also called BCRP, breast cancer resistance protein) is a 655 amino acid protein that has a molecular weight of 72 kDa and consists of an N-terminal nucleotide-binding domain and a six-segment C-terminal transmembrane domain, as well as a hydrophobic and an extracellular loop between TM5 and TM6 [148,149]. Having a membrane localization, protein is part of the ATP-binding cassette (ABC) drug transporter family and has the ability to use the energy formed by hydrolysis of ATP [150]. It contains a transmembrane binding domain and a nucleotide-binding domain and is activated by homodimerization. Some high-performance 3D structures of ABCG2 protein binding to different substrates and inhibitors indicate the molecular mechanisms of ABCG2 substrate selection, binding and transport [151]. Many ABCG2 substrates are additional substrates for other ABC transporters, P-glycoprotein (ABCB1), so the net effect of the availability of a dual-substrate drug (ABCB1/ABCG2) is attributed to combined action of two transporters [152]. ABCG2 actively recognizes and transports distinct molecules, often hydrophobic, many of which have a polycyclic structure or a relatively flat shape [153]. BCRP is an efflux transporter with an important role in detoxification by removing endo- and xenobiotics from many cell types (e.g., uric acid, steroid metabolites), in modulating drug absorption and having an important role in various pharmacokinetic stages. Similarly, it determines resistance to anticancer therapy [154,155,156,157,158,159]. In addition, a transporter reduces the transfer of therapeutic substances to tumor cells. ABCG2 also has an important role in the protection of stem cells. It has been shown that the number of tumorigenic stem cells is directly proportional to the progression of cancers, these cells being involved in the initiation of tumorigenesis, tumor angiogenesis, metastases, drug resistance and recurrences. Stem cells are characterized by increased drug and chemotherapeutic resistance due to their expression in the efflux pump of ABCG2 [160]. It is located on the interface of the blood-brain barrier (efflux capacity of the protein protects the central nervous system from compounds with endogenous toxicity), placenta, liver, kidneys, colon and small intestine [161,162,163,164,165,166,167]. The transporter is involved in decreasing the oral bioavailability, tissue distribution and effectiveness of many therapeutic agents [168]. ABCG2 transporters are known to be responsible for limited exposure of the brain to anticancer drugs, the effectiveness of which reduces it, especially in case of brain metastases [169]. The clinical significance of BCRP expression is correlated with the response to therapy and the prognosis of gastrointestinal, lung, breast, head and neck tumors, ovarian, prostate, glioblastomas and leukemias and so forth [164]. BCRP has important functions in excluding antitumor drugs from various cancer cells, being a significant therapeutic barrier. Since it has subspecies polyspecificity and is present in many tissues, protein is an important factor in resistance to therapy [170,171,172]. In some tumor formations, ABCG2 is strongly overexpressed, which is correlated with unfavorable clinical outcomes for these tumors [173]. Because the carrier is crucial for the pharmacokinetics of certain compounds, the US Food and Drug Administration and European Medicines Agency have indicated that pharmacokinetic studies and drug–drug interactions have been performed for it [174]. Functional characterization indicates that ABCG2 carries a large number of different substances, including methotrexate, mitoxantrone, SN-38 and various tyrosine kinase modulators (e.g., imatinib, nilotinib), which act as both substrates and inhibitors of ABCG2. Multidrug resistance proteins 1 and 2, which are encoded by ABCG2 genes, play an important role in the excretion of tyrosine kinase inhibitors. In addition, ABCG2 recognizes a wide range of positively or negatively modified substances and is resistant to most topoisomerases I or II, such as topotecan or doxorubicin, which is the reason for therapeutic failure [175,176,177,178,179]. ABCG2 polymorphism explains the low accumulation of anticancer agents (doxorubicin, tyrosine kinase inhibitors, adriamycin, platinum compounds, sorafenib and mitoxantrone) in cells and their altered chemotherapeutic response [180]. Previous studies have indicated the architecture of BCRP and the structural basis of the inhibition of this protein by small molecules and antibodies [82]. The inhibition of these transporters has a favorable impact on anticancer therapy by co-administering them with chemotherapeutic agents. There is a growing need to identify selective inhibitors that optimize drug absorption and prevent possible side effects. Currently, there are a limited number of ABCG2 inhibitors, which are used as pharmacological tools in studies to highlight pathological/physiological role of this transporter, to cross the blood-brain barrier of these biologically active molecules and to reduce resistance to polytherapy [181].

#### 2.1.3. Chalcones with Activity on ABCG2

Licochalcone A (LCA, Appendix A, Compound **9**, Figure 1), the most studied natural licochalcone, is present in roots of Glycyrrhiza inflata, has known pro-apoptotic and antiproliferative properties on different cell lines. Wu et al. determined the effect of LCA on multidrug-resistant ABCG2-overexpressing cancer cells. LCA has been shown to significantly alter the chemosensitivity of R482-HEK293 (ABCG2-transfected HEK293 human cells), S1-M1-80 (human colon cancer) and H460-MX20 (human non-small-cell lung cancer NSCLC) cells. For two known substrates of ABCG2 (mitoxantrone and topotecan), the activity was directly correlated with concentration [182,183].

Fan X et al. evaluated the inhibitory effects of some flavonoids on resistant proteins in breast cancer. The chalcones included in the study are LCA and licochalcone B, echinatin, isoliquiritin and isoliquiritigenin (Appendix A, Compounds **9**–**13**, Figure 2). A computational model (CDOCKER) was used to investigate molecular models of BCRP binding. The spatial conformation of BCRP-docked chalcones is different from that of mitoxantrone substrate. Mitoxantrone has a Pi-Alkyl potential with Val442 and two conventional bonds with Thr435 and Gln398. Analyzed chalcones have presumed potential Π-Π interactions with Phe439 and/or potential Pi-Alkyl interactions with Val546, which are essential for the strong inhibition of BCRP by flavonoids. Results show that the inhibitory potential of LCA and its analogue, isoliquiritigenin, is not correlated with conventional hydrogen bonds. Studies to identify essential pharmacophores responsible for inhibition and biological activity of chalcones on BCRP show that the basic elements are aromatic nuclei, hydrophobic groups and acceptors of hydrogen bonds [184].

A series of 44 chalcones and their analogues (Appendix A, Compounds **14**–**57**) were synthesized to evaluate the inhibitory potential and selectivity of ABCG2 by evaluating the effect on the transporter of mitoxantrone, a known substrate of ABCG2. Cell lines determined were ABCG2-transfected human fibroblast HEK293 (immortalized human embryonic kidney cells). Substituents on the two subunits (aldehyde and acetophenone) are the groups frequently present in natural compounds (methoxy and hydroxy groups), and their positions are 2,4 and 6 on acetophenone and 2, 3, 4, 5 and 6 on aldehyde. From the preliminary analysis, it was identified that -OR substituents are the most favorable for inhibitory activity. The substitution of aldehyde with chlorine in positions 2 or 3 also increases the inhibition of mitoxantrone efflux. In contrast, chlorine substitution at position 4 has a negative impact on ABCG2-mediated efflux inhibition [185]. The inhibitory activity of ABCG2 for the 44 chalcones was expressed as a percentage for the concentrations of 2 μM and 10 μΜ. For non-heterocyclic chalcones (Appendix A, Compounds **14**–**47**, Figure 3), the best inhibitory capacity is for compounds that are substituted on acetophenone with two methoxy groups at positions 2 and 4 and a hydroxy group at position 6 (Appendix A, Compounds **40**–**47**). On aldehyde, the position and number of methoxy groups are important. A substitution with -OMe in positions 2, 6 and 3, 5 (Appendix A, Compounds **18**, **21**, **22**, **24**, **26**, **27**, **31**, **34**, **36**, **37**, **41**, **45**, **46**) is the most favorable for inhibitory activity. A substitution with a single methoxy group at position 2 or 3 (Appendix A, Compounds **15**, **16**) is more favorable than the unsubstituted compound (Appendix A, Compound **14**). The activity of indole analogs is very different for the 1-indolyl and 3-indolyl series (Appendix A, Compounds **48**–**57**), which is explained by the impact of the electronic distribution in molecules on inhibitory activity. In a series of 1-indolylchalcones (Appendix A, Compounds **48**–**52**), heterocycle has a positive impact, the activity of these compounds being similar to that of the chalcones substituted on acetophenone with two methoxy groups in positions 2 and 4 and a hydroxy group in position 6 (Appendix A, Compounds **40**–**47**). Even though the basic structure is the same for all synthesized and analyzed compounds, there is a strong link between the structure of the compound and its activity. The positive contribution of the hydroxy group at position 6 of acetophenone is correlated with the electrostatic properties of these compounds (Appendix A, Compounds **40**, **41**, **42**). Similarly, the favorable contribution of the methoxy group on 5 position of aldehyde is explained (Appendix A, Compound **41**). On the other hand, the positive contribution of the methoxy group on 2 position of aldehyde (Appendix A, Compounds **40**, **50**, **51**) and the methoxy group at position 2 of acetophenone (Appendix A, Compounds **40**, **41**, **44**) is due to steric effects. The negative effect of the methoxy group on 4 position (Compound **45** versus Compound **40** and Compound **46** versus Compound **41**) is due to electrostatic properties. The remaining N-methyl-1-indolyl does not show particular interactions with favorable impact, indicating that hydrophobic interactions are essential for the binding capacity of compounds. In conclusion, it can be stated about chalcones and indolylphenylpropenones analyzed that they are potential inhibitors of ABCG2. From the 44 compounds, two representatives (Appendix A, Compounds **51**, **40**), which have a methoxy group on position 4 of aldehyde, are potential candidates for preclinical studies [185].

The fact that methoxy groups are favorable for the action of chalcones on ABCG2 was also performed for four naphthochalcones (Appendix A, Compounds **58**–**61**, Figure 4), which were synthesized to evaluate their effect on cancer cell growth on five different entities, using MTT analysis (2,5-diphenyl-2H-tetrazolium bromide). Compounds exhibit dose-dependent inhibitory activity. 3-Halophenyl chalcones (Compounds **58**–**60**) have, at micromolar concentrations, an inhibitory capacity that is twice lower than the trimethoxylated derivative (Compound **61**). Since chalcones substituted with two methoxy groups at positions 3, 4 inhibit BCRP-type ABC efflux transporters, it is normal for MCF-7/Topo cell lines to be approximately 10 times more sensitive to chalcone action compared to other cell lines. Halophenyl chalcones have IC50 values between 5 and 9 μM (Compounds **58**–**60**), and the trimethoxylated derivative is 2–3 times more active (Compound 61). Trimethoxynaphthochalcone (Compound **61**) inhibits both BCRP and P-gp transporters. Chalcone induces apoptosis through intrinsic pathways, and a Michael-type electrophilic system is essential for this activity [186].

Since the presence of a quinazoline residue and methoxy groups is known to be correlated with a strong inhibition of ABCG [187], they synthesized a series of 22 quinazoline chalcones (Appendix A, Compounds **62**–**83**) containing methoxy groups in the molecule. Initial results indicated the importance of substituents on the aromatic nucleus of aldehyde. Depending on acetophenone substituents, the unsubstituted aldehyde compounds are inactive. Among variants analyzed, the substitution of aldehyde with two methoxy groups in positions 3 and 4 (Compounds **64**, **66**, **69**, **74**, **76**, **80**) is favorable for ABCG2 inhibitory activity. The introduction of a quinazoline nucleus into molecules causes an increase in activity. Compounds from a series that have a meta-acryloyl phenyl residue on acetophenone (Compounds **62**–**65**) show an inhibitory effect on ABCG2 with IC50 values between 0.2 and 2 μΜ. Chalcone with three methoxy groups in positions 6 and 7 of quinazoline and in position 4 of aldehyde (Compound **65**) is the most active (IC50 = 0.32 μM). The substitution of acetophenone with two methoxy groups at positions 3 and 4 (Compounds **62**, **64**) causes a decrease in activity. In case of quinazolines substituted on 2 position of phenyl (Compounds **66**–**68**, Figure 5), the results obtained are different. Chalcone substituted with two methoxy groups in positions 3 and 4 of aldehyde (Compound **66**) is the most active, followed by the chalcone substituted on meta position with methoxy (Compound **68**). An analogue with methoxy group on para position (Compound **67**) is the least active. By introducing an additional methoxy group on 2 phenyl residue of compounds, (Compounds **69**–**71**) are obtained derivatives with similar activities to unsubstituted quinazoline chalcones (Compounds **62**–**65**).

In case of compounds with an acetophenone para-acryloylphenyl residue (Appendix A, Compounds **72**–**83**), the most active compounds are methoxy-substituted monosubstituted derivatives. The substituted derivatives in positions 6 and 7 of quinazoline with methoxy groups and monosubstituted ones are the least active (Compounds **72**–**75**). Compared to compounds from the first series (with a meta-acryloylphenyl residue), the para-substituted compounds have a lower activity, substitution on 2 position of phenyl on the heterocycle level being a favorable variant (Compound **62**-IC50 = 1.30 μM versus Compound **72**-IC50 = 0.82 μM and Compound **64**-IC50 = 1.71 μM versus Compound **74**-IC50 = 1.23 μM). Two derivatives substituted with methoxy groups at positions 6 and 7 of quinazoline and with a single methoxy group on aldehyde (Compounds **65**, **75**) have the best activity of the eight unsubstituted derivatives on 2 position of quinazoline. For quinazoline chalcones with a 2-phenyl residue (Compounds **76**–**78**), a similar effect to that of derivatives 66–68 was observed. From these, compound **76** has the best activity (IC50 = 0.29 μM), and the least active is the compound with a methoxy group on 4 position of aldehyde (Compound **78**, IC50 = 3.55 μM). Similar to meta-acryloyl compounds, the substitution of phenyl residue on quinazoline with two methoxy groups on 3 and 4 results in an increase in the activity of para-acryloyl derivatives as well. Compound **80** is the most active in the series (IC50 = 0.19 μM). Compound **81** (IC50 = 0.36 μM) is also a potent inhibitor of Ko143 (the most potent BCRP inhibitor known) and is the most active inhibitor of ABCG2 [188].

In another series of heterocyclic chalcones, in which inhibitory activity on ABCG2 was tested and obtained by Winter et al., 12 new compounds with a quinoxaline residue were synthesized (Appendix A, Compounds **84**–**95**). Depending on the t number and position of existing methoxy groups on the phenyl residue of acetophenone, chalcones have significant inhibitory effects on ABCG2. The best inhibitory properties have compounds substituted with two or three methoxy groups on acetophenone (Compounds **85**, **86**, **87**, **88**, **90**, **91**, **93**, **95**). Chalcone having a hydroxy group on 4 position of acetophenone has a much lower ability to inhibit ABCG2 (Compound **89**). Compared to chalcones in which the aldehyde subunit has a 3,4-methylenedioxyphenyl or 2-naphthyl residue, chalcones with a quinoxaline residue are much more active. Following evaluations, acetophenone substituents were found to have a significant influence on the ABCG2 inhibitory activity of chalcones. Substitution with chlorine, bromine, trifluoromethyl, nitro, cyano and hydroxy is not favorable for this activity [189].

Another study included 35 chalcones substituted in positions 2, 3 and 4 of acetophenone with a benzamide residue and in positions 3, 4 of aldehyde residue (especially with methoxy groups) (Appendix A, Compounds **96**–**131**), which were synthesized to evaluate their inhibitory potential on ABCG2 by Pheophorbide A and Hoechst 33342 methods, using MDCK II BCRP cell lines. Depending on the position of amide on the rest of acetophenone (ortho, meta or para), the series of compounds was divided into three groups. The first group includes compounds (Compounds **96**–**98**) in which amide is located on the meta position of acetophenone and is substituted with a phenyl, 4-nitrophenyl or 3-bromophenyl residues. The best activity has a phenyl-substituted amide compound (Compound **96**, IC50 = 2.18 μM in pheophorbid A). The 4-Nitrophenyl-substituted compound (Compound **97**) is not active. The second group has compounds (Compounds **99**–**115**) in which amide is located on the para position, and it is substituted with halogens, methoxy, cyano or nitro. With the exception of chlorine-substituted compounds (Compounds **105**–**107**), meta-substituted compounds are more active than para-substituted or disubstituted compounds. Of the para-benzamide compounds tested, the compound in which phenyl is not substituted on amide is the most active (Compound **99**, IC50 = 1.30 μM in pheophorbid A), and the least active compound is the one substituted with fluorine in 2 position of phenyl from amide (Compound **108**). Compounds with nitro or cyano groups on benzyl phenyl (Compounds **100**–**102**, **109**, **110**) have inhibitory activities on ABCG2 similar to halogen-substituted chalcones (Compounds **103**–**108**). The least active is a compound substituted with trifluoromethyl (Compound **110**, IC50 = 16.1 μM in pheophorbid A). In case of methoxylated derivatives of phenyl benzamide, the contribution of methoxy groups is additive. The most active are compounds with a methoxy group on 2 or 3 positions of phenyl (Compounds **111**, **112**), and the least active is the compound substituted in position 4 (Compound **113**). The third group of compounds contains an amide group on ortho position of acetophenone (Compounds **116**–**131**). Compared to derivatives from the para-substituted series, ortho-substituted chalcones are less active (except for compounds with nitro groups). The substitution of phenyl from the amide group with a chlorine causes a slight increase in their activity, this being the least active compound in the group (Compound **128**, IC50 = 0.77 μM in pheophorbid A). The 3-Methoxylated derivative (Compound **122**) has an activity similar to unsubstituted derivative. The replacement of phenyl with a 2-thienyl residue (Compound **127**) is favorable, as it is the most active compound. The derivative with a 3-quinolinyl residue (Compound **123**) is twice as potent as the one with a 3-pyridyl residue (Compound **125**). The replacement of 3,4-dimethoxy groups on aldehyde with chlorine or 3-methoxy-4-fluoro (Compounds **130**, **131**) resulted in a significant decrease in inhibitory capacity [190].

Solórzano et al. also synthesized six tariquidar-chalcones (Appendix A, Compounds **132**–**137**, Figure 6) in order to evaluate their cytotoxic potential, starting from the idea that tranquilizer analogues are selective inhibitors of ABCG2 susceptible to hydrolysis (which limits their use in biochemical and biological studies). Synthesized chalcones were investigated by Hoechst 33342 microplate analysis using MCF-7/Topo cells with overexpressed ABCG2. Standards used were fumitremorgin C and tariquidar. Chalcones have a maximum activity between 72 and 111%. Compounds with a methyl group on 4 position of aminochalcone (Compounds **135**–**137**) have a higher potency and a maximum inhibitory effect compared to unsubstituted compounds (Compounds **132**–**134**), indicating the importance of the substitution on 4 position of acetophenone. Their maximum effect is between 86 and 111%. A compound substituted with three methyl residues-a group on 4 position of acetophenone and two groups on tetrahydroisoquinoline (Compound **135**) has an inhibitory capacity similar to standard (fumitremorgin). The introduction of an ethylene or triethylene glycol residue (Compounds **133**–**134**, **136**–**137**) on tetraisoquinoline residue does not influence IC50 [191].

#### 2.1.4. Tubulin

Microtubules are cellular structures with a fundamental role in many essential biological processes, such as cytoskeletal architecture, intracellular transport, motility, chromosomal segregation and mitosis [92,192,193,194,195,196,197]. In the interface, microtubules are organized on cytoplasm in the form of a matrix-type network [198]. These are non-covalent mesoscopic polymers composed of α,β-tubulin protein heterodimers, which associate with and form protofilaments, which subsequently bind laterally and form microtubules. These dynamic structures are constantly elongated or shortened in all phases of cell cycle by adding or removing tubulin heterodimers from the ends of microtubules [199,200,201,202]. The assembly of protofilaments to microtubules is done spontaneously, with a number of protofilaments between 9 and 16, resulting in microtubules with different diameters [203]. It is known that the organization of microtubules changes significantly during the cell cycle. On the interface, microtubules form matrices in the cytoplasm and are relatively stable, and during mitosis, they form a bipolar mitotic spindle and become dynamic. When proliferative cells are exposed to microtubule inhibitors, bipolar spindle formation and the attachment of microtubules to kinetochores is inhibited, thus, activating the control point of division spindle assembly [204]. The stabilization of microtubules by acetylation is involved in cell migration [205]. Tubulin, in a depolymerized form, has the ability to influence cell physiology in various forms. For example, many microtubule-associated proteins have numerous sites of interaction with tubulin and microtubules. Thus, proteins containing the domain of overexpressed tumor genes use such multivalences to produce microtubule assembly. Mainly, tubulin, through its ability to influence the activity of proteins associated with microtubules, forms strong loops that connect the dynamics of microtubules to their basic matrix [206]. The dynamics of microtubules are strongly correlated with the regulation of cellular functions, such as intracellular transport and pathological processes [207]. This is essential for the optimal functioning of microtubules, in particular for the formation of dividing spindle in mitosis process. The disruption of this dynamic causes changes in cellular functions, influences the replication process and induces apoptosis. For these reasons, microtubules are one of the most studied targets of anticancer therapy. They actively participate in the formation of the centrosome, a formation characteristic of the G2/M phase of the cell cycle [208,209,210,211,212]. The unique feature of microtubule-binding agents, which is not present in other classes of anticancer agents, is their complexity and structural diversity, which determine many possibilities for optimization and modulation [213]. Compounds that interfere with cell division as a result of binding to α,β-dimers, oligomers or polymers have been extensively studied recently. Antimitotic agents, including different natural, synthetic or semi-synthetic products, have various chemical structures. Agents that interact with tubulin in the same binding region also belong to this category. Antimitotic agents include inhibitors of microtubule assembly (binds to Vinca domain or have sulfhydryl groups that alkylate tubulin) and microtubule-stabilizing agents (compounds that have a very high binding affinity, e.g., paclitaxel, docetaxyl, epothilone) [214,215,216,217]. Agents that target the field of colchicine (colchicine, podophyllotoxin, combrestatin A4) or those that bind to the domain of Vinca alkaloids (vincristine, vinblastine, vinorelbine) are defined as inhibitors of tubulin assembly as destabilizing agents [218,219,220,221]. 

Colchicine has a particular interest due to its antimitotic properties. The therapeutic potential of colchicine binding site has been investigated for its applications in chemotherapy. Colchicine binds strongly to the β subunit of tubulin. Cys241 residue forms hydrogen bonds with the trimethoxyphenyl residue of colchicine, and tubulin residues Thr179 and Val181 form hydrogen bonds with the troponelone residue of colchicine [218,222]. It is an increased need to identify new agents that bind to colchicine binding site. It is known that the intensity of the bond is dependent on the interaction of trimethoxyphenyl residue of colchicine with the binding site, and inhibition is achieved by interactions between oxygen atoms of the topolone network [223].

#### 2.1.5. Antitubulin Chalcones

Malik et al. evaluated the potential for the inhibition of microtubule polymerization by semisynthetic chalcones (Appendix A, Compound **138**, (E)-1-(2-(allyloxy)-4-methoxyphenyl)-3-(4-methoxyphenyl)prop-2-en-1-one, Figure 7). Assays were performed in vitro on HCT116 (human colorectal carcinoma) and MV-4-11 (human acute myeloid leukemia) cell lines. The derivative was observed to be a robust inducer of histone H3 to S10 phosphorylation after 8 h of treatment with 25 μM of the compound. The treatment of HCT116 or MV-4-11 cells with different concentrations of chalcone at 8 h resulted in a significant increase in histone H3 phosphorylation at S10, even at concentrations of 0.78 μM. Chalcone-induced cell cycle blockade was also confirmed by a FACS analysis of MV-4-11 cells. Through biochemical tests performed, it was observed that the derivative inhibits polymerization of microtubules depending on concentration (2–25 μM). Chalcone docking studies indicate its binding to a colchicine binding site, one of the aromatic residues being placed on trimethoxyphenyl test region of colchicine. The compound forms two hydrogen bonds with tubulin on Leu255 and Cys241 residues. The phenyl residue adjacent to the carbonyl group forms Π-σ interactions with Leu248, and the other aromatic residue interacts with residues Ala180 and Lys254 [224]. 

Millepachine (Appendix A, Compound **6**), a chalcone extracted from seeds of Milletia pachycarpa, has in vitro anticancer properties. It is known that flavonoids act by mechanisms such as cell cycle blocking by inhibiting cyclin-dependent kinase and by inhibiting tubulin polymerization. Hybrid molecule of millepachine with indole (Appendix A, Compound **139**) also inhibits the polymerization of tubulin on hepatocellular cancer cell lines 3.6 times stronger than natural chalcone. An increase in activity can be attributed to the introduction of indole residue into molecules, this being a promoter for the binding of the compound on the tubulin cavity. Similarly, millepachine, indole derivative distorts dynamics of the intracellular tubulin-microtubule system and, implicitly, blocks cell cycle by blocking their G2/M phase. Two compounds (Appendix A, Compounds **140**, **141**) were used to evaluate the class of antitubulin agents (stabilizers or destabilizers of tubulin) to identify in which chalcones belong. The amount of each compound used for microtubule dynamics methods is 15 μM with paclitaxel and combrestatin A-4 as controls. Paclitaxel (10 μM) stimulates tubulin polymerization, and combrestatin inhibits it. For the chalcones analyzed, a marked inhibition of polymerization was observed, with data showing that these compounds have the ability to effectively inhibit tubulin polymerization and act as destabilizing agents. To determine the possible binding of derivatives, docking studies were performed to demonstrate that these compounds have the same binding site as colchicine and 4-hydroxymillepachine. Binding patterns were generated for four compounds (Compounds **140**, **141**, colchicine and 4-hydroxymillepachine) on the colchicine binding site. The surflex docking score is 12.53 for compound **140** (€-2-(2-methoxy-5-(3-(5-methoxy-2,2-dimethyl-2H-chromen-8-yl)-3-oxoprop-1-en-1-yl)phenoxy)-N-(2-methoxyphenyl)acetamide) and 12.31 for compound **141** ((E)-N-(2-chlorophenyl)-2-(2-methoxy-5-(3-oxo-3-(3,4,5-trimethoxyphenyl)prop-1-en-1-yl)phenoxy)acetamide); a higher score indicating a higher affinity for the mode of interaction of compound **141**. In the binding site, compound **140** is surrounded by Cys241, Leu248, Lys352 and Leu352. In particular, compound **140** forms three hydrogen bonds with polar amino acid Asn101, indicating a possible strong electrostatic interaction with protein. In addition, the hydrophobic residue of compound **140** is embedded in a pocket that interacts with several hydrophobic residues that facilitate strong tubulin binding. Compound **141** binds similarly to colchicine. In this case, docking studies showed that the 3,4,5-trimethoxyphenyl residue of compound **141**, similar to colchicine, is compatible with a hydrophobic site and adopts an energetically stable conformation. In addition, the methoxy group and carbonyl oxygen of t 3,4,5-trimethoxyphenyl residue and the oxygen atom of amide from compound **141** act as acceptors and establish three hydrogen bonds with Ser178, Tyr224 and Asn249, which are consistent with the observation that tubulin heterodimer is stabilized by colchicine and confirms that this subunit is essential for binding. Similarly, an interaction with Ser178 or Tyr224 in the 4-hydroxymillepachine can be observed. Essential electrostatic interactions between methoxy groups of 3,4,5-trimethoxyphenyl residue and Ser178, Lys254 and Asn101 residues of the neighboring α subunit are observed on a binding pocket, demonstrating a plausible competitive mechanism of action on the colchicine binding site [130].

Another study was developed by Wang G et al., who synthesized a series of naphthalenchalcones (Appendix A, Compounds **142**–**161**, Figure 8) in order to evaluate their ability to inhibit tubulin polymerization. To evaluate the possibility that the representatives of this class have an action on the microtubule system, compound (E)-3-(3-hydroxy-4-methoxyphenyl)-1-(2-methoxynaphthalen-1-yl)prop-2-en-1-one (Compound **142**), one of the most active compounds from the series, has been investigated for its ability to inhibit tubulin polymerization in a direct correlation with its concentration. This is a precise indicator of its action by interfering with the polymerization of microtubules. Treatment with 3, 6, 12.5 and 25 μM of the compound inhibited tubulin polymerization by 21, 41, 60 and 82%. Compound **142** is more active than the control compound, colchicine (IC50 value of compound **142** = 8.4 μM, IC50 value of colchicine = 10.6 μM). These results show that derivative 142 is an inhibitor of tubulin polymerization, which binds to tubulin and influences polymerization of microtubules. The determination of tubulin polymerization inhibitory activity for compound **142** was followed by an investigation of the cellular mechanism of action on MCF-7 cancer cells, using cytometric analysis. To elucidate the molecular mechanism of the derivative, the effect on cell cycle progression in MCF-7 cells was initially studied. The control group has typical cell cycle characteristics in G1, S and G2/M phases. In contrast, after treatment of MCF-7 cells with 2 μM of compound **142**, the accumulation of cancer cells was detected in the G2/M phase with an intensity 5.5 times higher than in the control group (84.55% for the treated group and 15.19% for the control group). The results indicate the potential of chalcone to block the cell cycle in the G2/M phase and to stop cell mitosis, thus, inhibiting the proliferation of MCF-7 cells. To explain how to bind this class of tubulin compounds, a docking study was performed for chalcone 142 which is based on the binding pocket of tubulin colchicine. The results estimate a binding energy of −8.8 kcal/mol. The compound adopts “L-shaped” conformation on tubulin pocket. The 2-Methoxynaphtyl group of the compound is located in the hydrophobic pocket, is surrounded by residues Cys241, Leu248, Ala250, Leu252 and Leu255 and forms a strong hydrophobic bond. The detailed analysis shows that phenyl residue from the center of compound **142** forms a cation-Π-type interaction with Lys254 residue. Gln11, Leu248 and Leu 255 residues form interactions through three hydrogen bonds with compound **142** [225].

A new series of chalcone derivatives with a diaryl ether residue (Appendix A, Compounds **162**–**177**, Figure 9) were modulated and synthesized in order to evaluate their activity on tubulin polymerization. Of the compounds obtained, chalcone substituted with a methoxy group on an aromatic ring of aldehyde (compound **163**, (E)-3-(4-methoxyphenyl)-1-(3-(3,4,5-trimethoxyphenoxy)phenyl)prop-2-en-1-one) is the most active on MCF-7, HepG 2 (hepatoblastoma-derived cell line) and HTC116 cancer lines. Tubulin inhibition activity for chalcone 163 was evaluated in vitro, using colchicine as a standard. Tubulin protein was mixed with different concentrations of compound **163** (0.8 μM, 1.5 μM, 3.0 μM, 6.0 μM, 12.5 μM and 25 μM). In case of the inhibition of tubulin protein by derivative 163, a tendency to decrease the intensity of fluorescence similar to colchicine was observed. IC50 values for compound **163** and colchicine are 20 μM and 10.6 μΜ. The cytometric method using MCF-7 cells was used to examine the cytotoxic effect of chalcone (2.5 μM and 5.0 μM). Cells from the control group have 59.61% of cells in phase G1, 18.11% in phase G2 and 22.27% on phase S of the cell cycle. Treatment with 2.5 μM of the compound did not significantly influence the cell cycle. The use of 5.0 μΜ of the compound greatly increased the percentage in G2/M phase cells (50.64%). This result shows that the derivative has increased antiproliferative activity on MCF-7 cells by increasing the percentage of cells in the G2/M phase. Docking studies suggest that derivative interacts and binds to the binding site of colchicine at tubulin level [226].

Du et al. analyzed a new indole chalcone, (3-6-methoxy-2-methyl-1H-indol-3-yl)-1-pyridyl-2-propen-1-one) (Appendix A, Compound **178**), in order to identify the mechanism by which it induces cell death and to assess whether it crosses the blood-brain barrier. The analyzed chalcone acts as a destabilizing agent for microtubules and induces mitosis suppression and programmed cell death depending on the activity of caspases on glioblastoma and on various cancer cell lines. The activity of compound **178** is correlated with its ability to bind to the binding site of β-tubulin colchicine [227]. Other indole-chalcone derivatives were analyzed from the point of view of anti-tumor activity, related to tubulin and TrxR [228].

Another study by Canela et al. comprises a series of chalcones with a dioxolane residue in acetophenone (Appendix A, Compounds **179**–**187**, Figure 10), given that this substitution is frequently present in natural colchicine binding ligands. Using impedance cell growth monitoring, a time- and dose-dependent antimitotic effect was identified on MDA-MB-231 (human breast adenocarcinoma) cancer cells treated with (E)-3-(3“-amino-4”-methoxyphenyl)-1-(5’-Methoxy-3’,4’-methylenedioxyphenyl)-2-methylpropen-2-en-1-one (Compound **185**), the most active compound from the series. At concentrations greater than 5 nM, the compound causes a rapid and continuous decrease in cell index, indicating a reduction in cell adhesion and/or toxicity. Using low concentrations of the compound, a decrease in the cellular index was observed at the first 14 h and a restoration occurred at 24–48 h, with the properties being similar to those of compounds that act on tubulin. Flow cytometry indicates an accumulation of cells in the G2/M phase 24 h after treatment with 10 nM compound. The induction of apoptosis was confirmed by measuring the translocation of phosphatidylserine from the cytoplasmic medium to the extracellular medium and by the activation of caspase 3 by the compound. The evaluation of the antitubulin activity of compound **185** on MDA-MB-231 cell cultures (human breast adenocarcinoma) shows the derivative blocks mitosis. Using flow cytometry, an accumulation of cells in the G2/M phase after 24 h of treatment with 10 nM of the compound was observed. Chalcone also causes a dose-dependent increase in subG1 phase cells and a sub-diploid DNA content, which is characteristic of apoptotic cells (45% ± 9 and 19% ± 8 to 10 and 1 nM of the compound). The induction of apoptosis was confirmed by measuring the translocation of phosphatidylserine from the cytoplasm into the extracellular environment of the cytoplasmic membrane and by activating caspase 3 by the compound [229].

Lindamulage et al. synthesized 24 quinoline chalcones to evaluate their anticancer activity. Two chalcones of series, (E)-3-(3-(2-methoxyphenyl)-3-(oxoprop-1-enyl)quinolin-2(1H)-one (Appendix A, Compound **188**) and (E)-6-methoxy-3(3-(2-methoxyphenyl)-3-oxoprop-1-enyl)quinolin-2(1H)-one (Appendix A, Compound **189**) show increased efficiency and selectivity of the cells. Both chalcones bind to the colchicine binding site and cause a prolongation of mitosis by acting on the dividing spindle, leading to cell death. Compounds destroy tumor cells with overexpressed MDR1 and MRP1, which are resistant to colchicine, ABT and paclitaxel. Chalcones 188 and 189 have a strong synergistic effect on tumor cells, including pluritherapy-resistant tumors. Since the analyzed quinazoline derivatives block the functioning of microtubules, the activity of polymerizing microtubules in the absence and presence of chalcones was examined. In vitro studies have shown that the activity of chalcones on microtubules is similar to the activity of nocodozole, but different from that of paclitaxel. This indicates the link between the action of chalcones and the polymerization of microtubules. Studies of intrinsic fluorescence of tryptophan, a method frequently used to determine the binding affinity of compounds to tubulin heterodimers, indicate the possibility of compounds to bind to the binding site of colchicine [230]. 

A series of 25 heterocyclic chalcones (Appendix A, Compounds **190**–**214**, Figure 11) were synthesized to evaluate their cytotoxic potential in vitro. After evaluating the interaction capacity between synthesized compounds and microtubules, it was shown that compound **190** had the capacity to inhibit tubulin polymerization in vitro. Chalcone interrupts cell cycle in the G2/M phase, an indicator of tubulin polymerization. The ability of the compound to inhibit tubulin polymerization was determined by combrestatin A4 and paclitaxel. Experimental data show that the hybrid molecule is a potent inhibitor of tubulin polymerization, having IC50 values = 9.66 ± 0.06 μM [231].

Qiu et al. also synthesized a series of the shikonium-derived chalcones (Appendix A, Compounds **215**–**232**, Figure 12) to evaluate their potential to inhibit tubulin polymerization by in vitro assays. IC50 values are similar for inhibiting tubulin polymerization and for cellular antiproliferation testing. Among the synthesized compounds, compound **226** has the best anti-tubulin activities, having IC50 values = 2.98 ± 0.53 μM. Studies of mechanisms of action show that derivative 226 can induce apoptosis of MCF-7 cells, reduce the potential of mitochondrial membranes and cause an accumulation of cells in the G2/M phase of the cell cycle, and the effect is to disrupt the microtubule system similarly to standard compound, colchicine. Of the compounds analyzed, derivative 226 has the best energy, the value of this compound being −64.6074 kcal/mol. The compound binds to the binding site of colchicine on tubulin through three hydrogen bonds with Ser178, Tyr202 and Lys254 residues. Data from in silico analysis show that derivative 224 is a potent inhibitor of tubulin polymerization [232].

Starting from the anticancer activity of natural and synthetic chalcones, Yan W et al. synthesized a 1,2,3-triazole hybrid of chalcones (Appendix A, Compound **233**) and tested the anticancer activity of compounds on liver cells lines. (E)-1-(4-((1-allyl-1H-1,2,3-triazol-4-yl)methoxyphenyl)-3-(2,4-dichlorophenyl)propen-2-en-1-one (Compound **233**) shows inhibitory values IC50 = 2.34 μM against tubulin polymerization, a value that indicates the potential of the compound to be a new candidate for novel antitubulin agents [233].

## 3. Conclusions

ABCG2 and tubulin are mechanisms by which cancer cells resist treatment. ABCG2 (also called BCRP) decreases the transfer of drugs to cancer cells. It is also sought to find substances that inhibit the formation of tubulin assembly under the influence of tumor cells, through the binding of the compounds to the colchicine binding site. Compound **41** and compound **80** have very good anticancer activity against ABCG2, having an IC50 of 0.17 µM and 0.19 µM (Figure 13), respectively, due to the methoxy substituents on the aromatic nucleus of the acetophenone residue, and the substitution of the phenyl residue by a heterocycle (quinazoline) with two methoxy groups. For the second mechanism, compounds **8** and **185** have an IC50 of 0.003 µM and 0.0039 µM (Figure 13), respectively. This strong activity is explained by the substitution of methoxy groups at the acetophenonic residue but also by the presence of a heterocycle (indole) in the molecule of compound **8**. In compound **185**, the activity is determined by the condensation of dioxolane with the phenyl of the acetophenonic group and by the existence of methoxy and amino substituents in the molecule.

## 4. Methods

The articles were selected from the PubMed and Google Scholar databases, taking into account the most representative articles about cancer and flavonoids, and about natural, semi-synthetic and synthetic chalcones with anticancer activity, with an emphasis on the mechanisms of ATP binding cassette subfamily G and tubulin. The inclusion criterion was that the articles were from the period 2010–2022, and the exclusion criterion was other mechanisms of anticancer action, different from ABCG2 and tubulin.

## Figures and Tables

**Figure 1 ijms-23-11595-f001:**
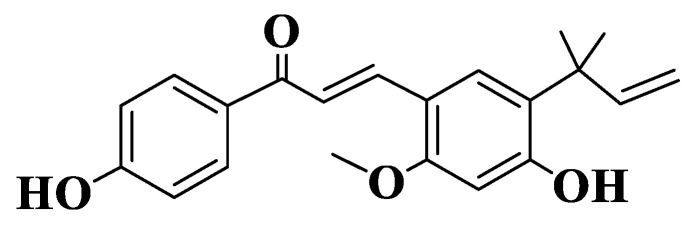
Structure of Licochalcone A.

**Figure 2 ijms-23-11595-f002:**
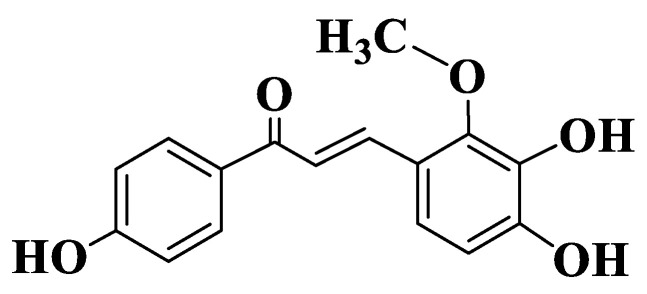
Structure of Licochalcone B.

**Figure 3 ijms-23-11595-f003:**
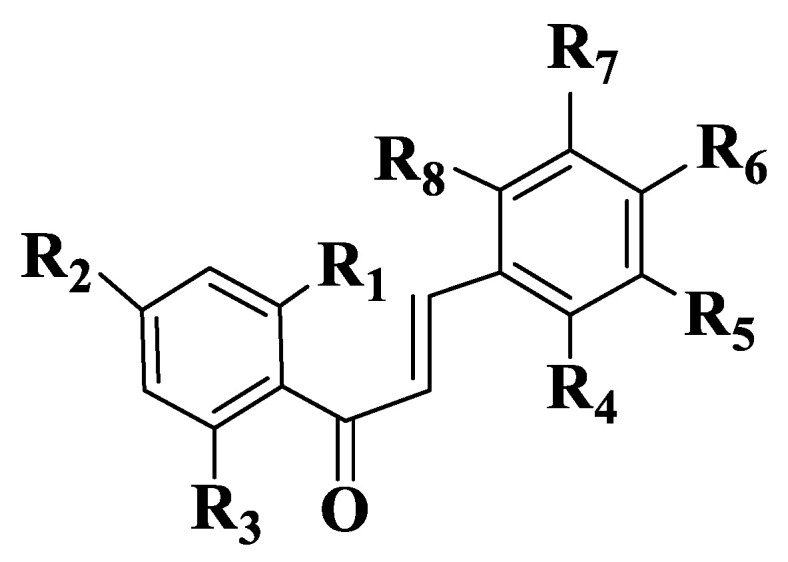
General structure of chalcones **14**–**47**.

**Figure 4 ijms-23-11595-f004:**
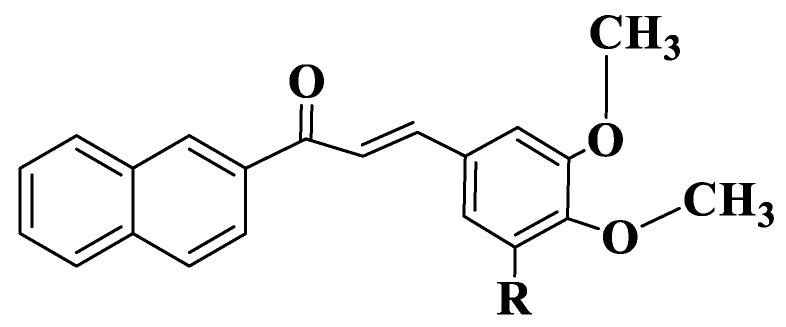
General structure of naphthochalcones (Compounds **58**–**61**).

**Figure 5 ijms-23-11595-f005:**
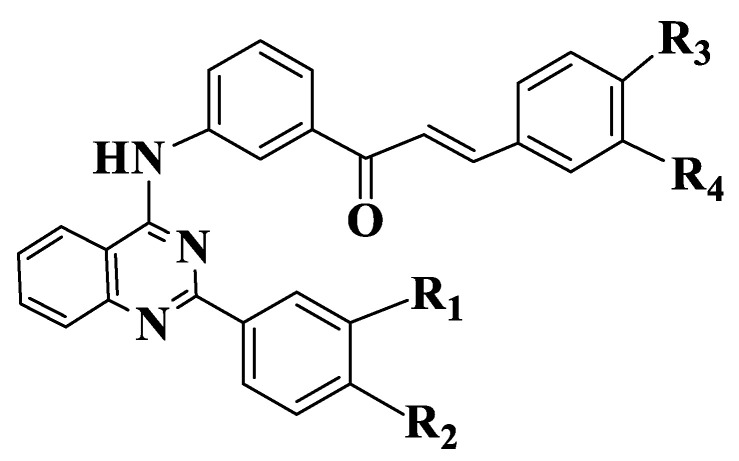
General structure of quinazoline chalcones (Compounds **66**–**71**).

**Figure 6 ijms-23-11595-f006:**
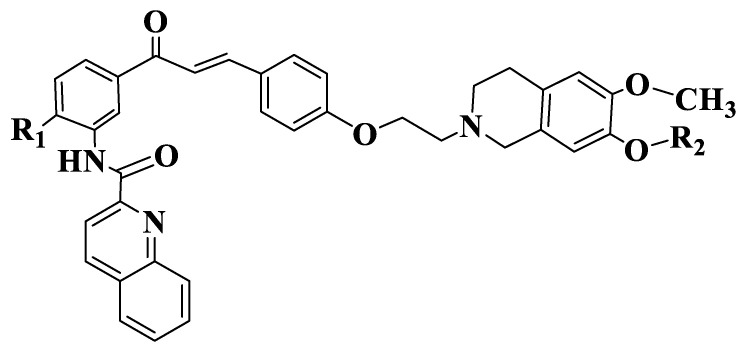
General structure of tariquidar-chalcones (Compounds **132**–**137**).

**Figure 7 ijms-23-11595-f007:**
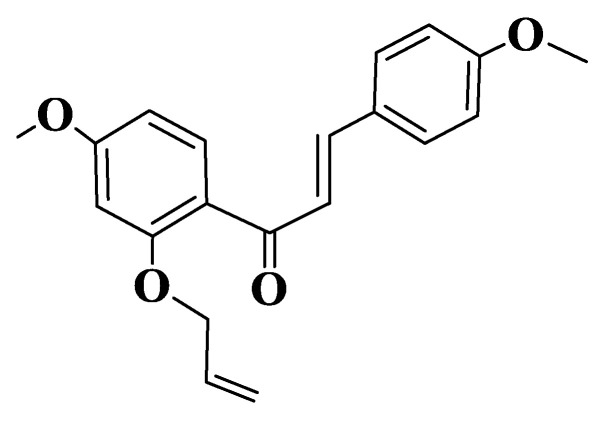
Structure of a semisynthetic chalcone (Compound **138**).

**Figure 8 ijms-23-11595-f008:**
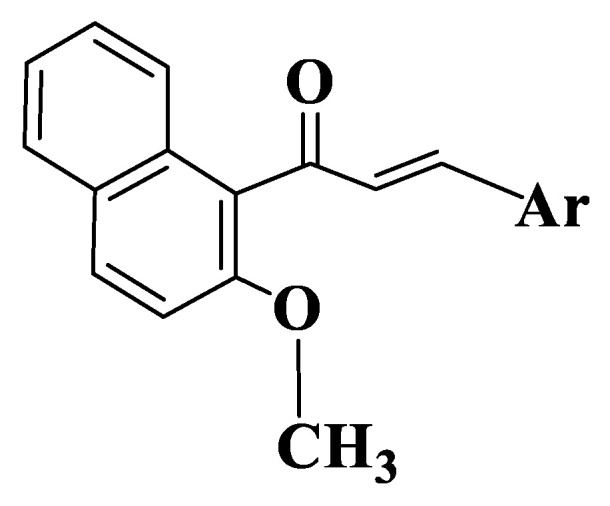
General structure of naphthalenchalcones (Compounds **142**–**161**).

**Figure 9 ijms-23-11595-f009:**
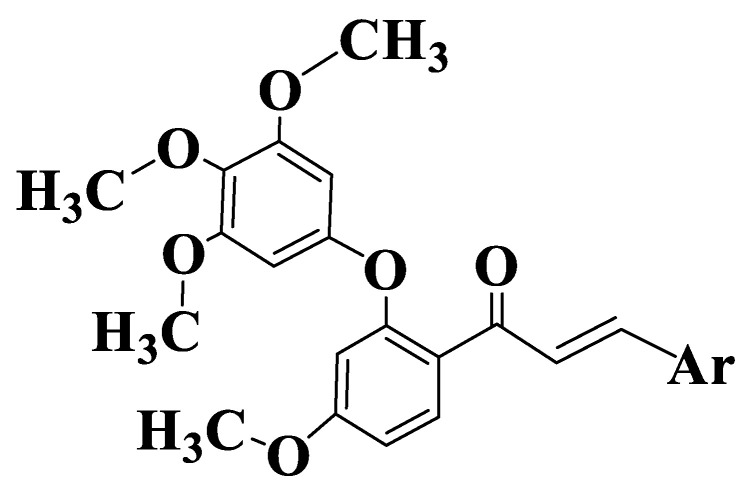
General structure of chalcones **162**–**177**.

**Figure 10 ijms-23-11595-f010:**
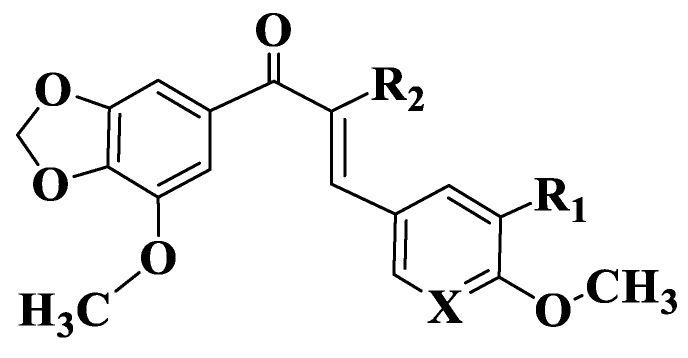
General structure of chalcones **179**–**187**.

**Figure 11 ijms-23-11595-f011:**
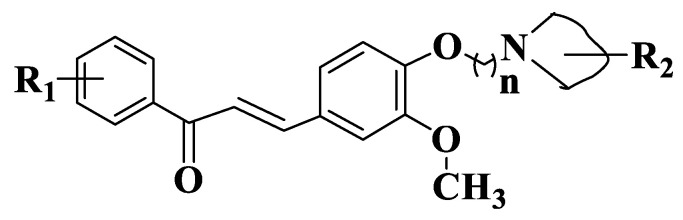
General structure of quinoline chalcones (Compounds **190**–**214**).

**Figure 12 ijms-23-11595-f012:**
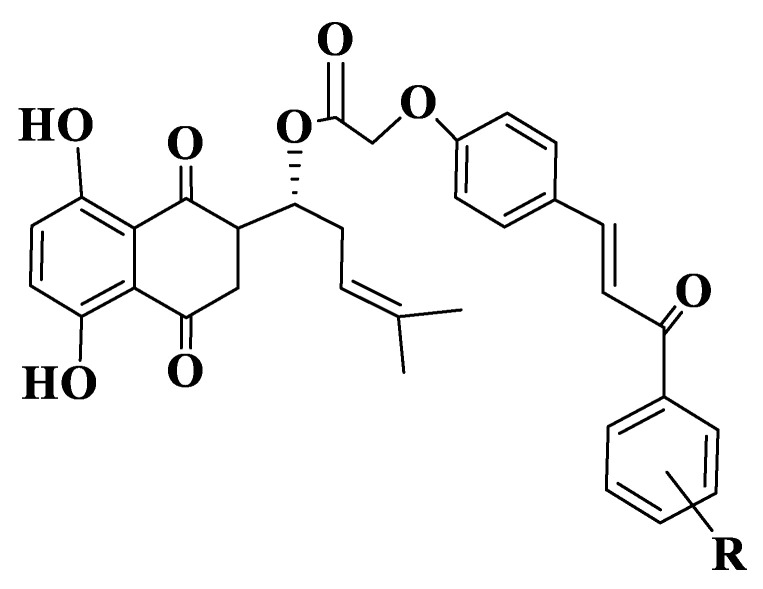
General structure of shikonium-derived chalcones (Compounds **215**–**232**).

**Figure 13 ijms-23-11595-f013:**
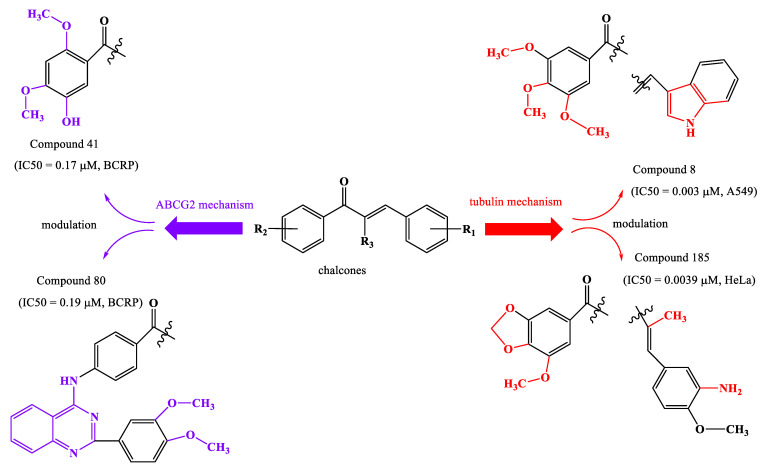
The two mechanisms and most active compounds.

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
