# Peer review of "Two Important Anticancer Mechanisms of Natural and Synthetic Chalcones"

_ijms, 2022, doi:10.3390/ijms231911595_

Round 1

Reviewer 1 Report

The review “Two Important Anticancer Mechanisms of Natural and Synthetic Chalcones” is well written and systematically describes recent achievements in the study of natural and synthetic chalcones as anticancer ABCG2 casette and tubulin inhibitors. I would like to suggest the following minor revisions: 1) to move Fig 1 and 2 with well-known data to the Support file and, in contract, to add some lead structures from Support file to the  Results to the main text; 2) to add a new Figure with an illustration and  summarizing   the main topic of the review. 

Author Response

Hello!

Thank You very much for Your amiability and work concerning our manuscript, and we are grateful all reviewers’ comments on the manuscript. These comments are all very valuable and helpful in increasing the quality of our manuscript. We have studied these comments carefully and made correction in the revised manuscript according to all reviewers’ comments. These changes are performed with Track Changes (MS Word 2013) in the manuscript and in the Supplementary Materials. The detailed answers point by point are presented below:

Question 1: To move Fig 1 and 2 with well-known data to the Support file and, in contract, to add some lead structures from Support file to the  Results to the main text;

Answer: Thanks for your helpful suggestion. I move Figure 1 and Figure 2 to the Supplementary Materials-Annex A, I have inserted in the text on page 2 and page 3 Figure A- Annex A (old Figure 1) and reference Constantinescu, T.; Lungu, C.N. Anticancer Activity of Natural and Synthetic Chalcones. Int. J. Mol. Sci. 2021, 22, 11306. I inserted in the text on page 20 Figure B- Annex A (old Figure 2) and reference Constantinescu, T.; Lungu, C.N. Anticancer Activity of Natural and Synthetic Chalcones. Int. J. Mol. Sci. 2021, 22, 11306.

I have added:

Figure 1. Structure of Hesperidin (page 3);

Figure 2. Structure of Xanthohumol (XN) (page 4);

Figure 3. Structure of Licochalcone A (page 6);

Figure 4. Structure of Licochalcone B (page 6);

Figure 5. General structure of chalcones 14-47 (page 7);

Figure 6. General structure of naphthochalcones (compounds 58-61) (page 8);

Figure 7. General structure of quinazoline chalcones (compounds 66-71) (page 8);

Figure 8. General structure of tariquidar-chalcones (compounds 132-137) (page 10);

Figure 9. Structure of a semisynthetic chalcone (compound 138) (page 12);

Figure 10. General structure of naphthalenchalcones (compounds 142-161) (page 13);

Figure 11. General structure of chalcones 162-177 (page 14);

Figure 12. General structure of chalcones 179-187 (page 15);

Figure 13. General structure of quinoline chalcones (compounds 190-214) (page 15);

Figure 14. General structure of shikonium-derived chalcones (compounds 215-232) (page 16);

Figure 15. The two mechanisms and most active compounds (page 17)

Question 2: to add a new Figure with an illustration and summarizing  the main topic of the review.

Answer: Thanks for your helpful suggestion. I have added a diagram for summarization: Figure 15. The two mechanisms and most active compounds.

In addition, we made some changes in the manuscript using Track Changes option from MS Word 2013, which we list further:

I corrected the general structure of compounds 162-177 (there were 2 small errors) and the general structure of compounds 190-214 (one small error).

For Table S1, S2 and S3 (from Supplementary Materials) I have inserted page numbers.

There was a problem with references 138-140. I have corrected now the reference 138-Jain in reference 140, the reference 139-Urbonavicius in reference 138, and the reference 140-Bala in reference 139 in the subchapter references.

On page 5 in the text, I corrected 142-145 in 142-146, that is, I added the reference 146-Constantinescu.

Since I inserted the reference 146-Constantinescu in the text in Figure A and Figure B, the reference number changed from 146-Constantinescu to 66-Constantinescu.

We appreciate for Editors/Reviewers’ warm work seriously, and hope that the correction will meet with approval.

Yours sincerely,

Alin Mihis

Reviewer 2 Report

Naturally occurring chalcones have been used in traditional medicine for many years; nevertheless, recent scientific advances have shown that both these molecules and their synthetic derivatives have a wide range of anticancer activities, including inhibition of tubulin and ABCG2. This article is a review of natural and synthetic chalcones which target tubulin and ABCG2 casette. The authors have performed an in-depth search of chalcones, as corroborated by an extensive bibliography. The authors describe the chalcones' mode of action against these two tumor targets, indicating the essential structural features that enable their successful inhibition. The review is written well. There are some modified parts and figures which resemble very much the article "Anticancer Activity of Natural and Synthetic Chalcones" by Teodora Constantinescu and Claudiu N. Lungu. Nevertheless, I do not see any plagiarism. I recommend this article for publication with minor corrections.

- It is strongly recommended to reformulate the last sentence in the abstract. 

- "Photo-Fries rearrangement" on p. 4 should be "photo-Fries rearrangement".

- "Flawokawain" should be "Flavokawain" in Supporting Information.

Author Response

Hello!

Thank You very much for Your amiability and work concerning our manuscript, and we are grateful all reviewers’ comments on the manuscript. These comments are all very valuable and helpful in increasing the quality of our manuscript. We have studied these comments carefully and made correction in the revised manuscript according to all reviewers’ comments. These changes are performed with Track Changes (MS Word 2013) in the manuscript and in the Supplementary Materials. The detailed answers point by point are presented below:

Question 1:

It is strongly recommended to reformulate the last sentence in the abstract.

Answer: Thanks for your helpful suggestion. I have rewritten the Abstract, I have modified ABCG2 in ATP binding cassette subfamily G.

Question 2:

- "Photo-Fries rearrangement" on p. 4 should be "photo-Fries rearrangement".

Answer: Thanks for your helpful suggestion. I have corrected the name "Photo-Fries rearrangement" on page 4 in "photo-Fries rearrangement".

Question 3:

- "Flawokawain" should be "Flavokawain" in Supporting Information.

Answer: Thanks for your helpful suggestion. I have corrected the name "Flawokawain" in "Flavokawain" in Supplementary Materials-Table S1.

At the recommendation of the expert reviewers, I made the following changes:

I have added:

Figure 1. Structure of Hesperidin (page 3);

Figure 2. Structure of Xanthohumol (XN) (page 4);

Figure 3. Structure of Licochalcone A (page 6);

Figure 4. Structure of Licochalcone B (page 6);

Figure 5. General structure of chalcones 14-47 (page 7);

Figure 6. General structure of naphthochalcones (compounds 58-61) (page 8);

Figure 7. General structure of quinazoline chalcones (compounds 66-71) (page 8);

Figure 8. General structure of tariquidar-chalcones (compounds 132-137) (page 10);

Figure 9. Structure of a semisynthetic chalcone (compound 138) (page 12);

Figure 10. General structure of naphthalenchalcones (compounds 142-161) (page 13);

Figure 11. General structure of chalcones 162-177 (page 14);

Figure 12. General structure of chalcones 179-187 (page 15);

Figure 13. General structure of quinoline chalcones (compounds 190-214) (page 15);

Figure 14. General structure of shikonium-derived chalcones (compounds 215-232) (page 16);

Figure 15. The two mechanisms and most active compounds (page 17)

In addition, we made some changes in the manuscript using Track Changes option from MS Word 2013, which we list further:

I corrected the general structure of compounds 162-177 (there were 2 small errors) and the general structure of compounds 190-214 (one small error).

For Table S1, S2 and S3 (from Supplementary Materials) I have inserted page numbers.

There was a problem with references 138-140. I have corrected now the reference 138-Jain in reference 140, the reference 139-Urbonavicius in reference 138, and the reference 140-Bala in reference 139 in the subchapter references.

On page 5 in the text, I corrected 142-145 in 142-146, that is, I added the reference 146-Constantinescu.

Since I inserted the reference 146-Constantinescu in the text in Figure A and Figure B, the reference number changed from 146-Constantinescu to 66-Constantinescu.

We appreciate for Editors/Reviewers’ warm work seriously, and hope that the correction will meet with approval.

Yours sincerely,

Alin Mihis

Reviewer 3 Report

In the review authors described two mechanisms of anticancer action of natural and synthetic chalcones that are or could be used as therapeutic agents in cancer treatment. The authors selected to describe two sites of action in potential anticancer treatment - ABCG2 (breast cancer protein) casette and tubulin. For each site of action the authors gave examples of chalcone (natural and/or synthetic) anticancer activity together with modes of action for active compounds. Selected compounds and their activity are described in sufficient details and present good examples of chalcones anticancer bioactivity.

The manuscript is well organized and the selected data is presented in a good and clear manner.

The drawn conclusions are appropriate and supported by the data presented in the review.

The references used in manuscript are well selected and they are relevant and sufficient.

The manuscript is well written and easy to read and does need only minor corrections:

Line 154 – please give a reference for prenylchalcone anticancer activity for hepatocellular carcinoma and prostate cancer mentioned in the text.

Please correct everywhere in the text et al. because somewhere it is with punctuation and somewhere without it.  

Author Response

Hello!

Thank You very much for Your amiability and work concerning our manuscript, and we are grateful all reviewers’ comments on the manuscript. These comments are all very valuable and helpful in increasing the quality of our manuscript. We have studied these comments carefully and made correction in the revised manuscript according to all reviewers’ comments. These changes are performed with Track Changes (MS Word 2013) in the manuscript and in the Supplementary Materials. The detailed answers point by point are presented below:

Question 1:

Line 154 – please give a reference for prenylchalcone anticancer activity for hepatocellular carcinoma and prostate cancer mentioned in the text.

Answer: Thanks for your helpful suggestion. I have added a new reference, 131- Harish, V.; Haque, E.; Åšmiech, M.; Taniguchi, H.; Jamieson, S.; Tewari, D.; Bishayee, A. Xanthohumol for Human Malignancies: Chemistry, Pharmacokinetics and Molecular Targets. Int. J. Mol. Sci. 2021, 22, 4478.

Question 2:

Please correct everywhere in the text et al. because somewhere it is with punctuation and somewhere without it. 

Answer: Thanks for your helpful suggestion. I have corrected in the text "et al" in "et al.".  

At the recommendation of the expert reviewers, I made the following changes:

. I move Figure 1 and Figure 2 to the Supplementary Materials-Annex A, I have inserted in the text on page 2 and page 3 Figure A- Annex A (old Figure 1) and reference Constantinescu, T.; Lungu, C.N. Anticancer Activity of Natural and Synthetic Chalcones. Int. J. Mol. Sci. 2021, 22, 11306. I inserted in the text on page 20 Figure B- Annex A (old Figure 2) and reference Constantinescu, T.; Lungu, C.N. Anticancer Activity of Natural and Synthetic Chalcones. Int. J. Mol. Sci. 2021, 22, 11306.

I have added:

Figure 1. Structure of Hesperidin (page 3);

Figure 2. Structure of Xanthohumol (XN) (page 4);

Figure 3. Structure of Licochalcone A (page 6);

Figure 4. Structure of Licochalcone B (page 6);

Figure 5. General structure of chalcones 14-47 (page 7);

Figure 6. General structure of naphthochalcones (compounds 58-61) (page 8);

Figure 7. General structure of quinazoline chalcones (compounds 66-71) (page 8);

Figure 8. General structure of tariquidar-chalcones (compounds 132-137) (page 10);

Figure 9. Structure of a semisynthetic chalcone (compound 138) (page 12);

Figure 10. General structure of naphthalenchalcones (compounds 142-161) (page 13);

Figure 11. General structure of chalcones 162-177 (page 14);

Figure 12. General structure of chalcones 179-187 (page 15);

Figure 13. General structure of quinoline chalcones (compounds 190-214) (page 15);

Figure 14. General structure of shikonium-derived chalcones (compounds 215-232) (page 16);

Figure 15. The two mechanisms and most active compounds (page 17)

In addition, we made some changes in the manuscript using Track Changes option from MS Word 2013, which we list further:

I corrected the general structure of compounds 162-177 (there were 2 small errors) and the general structure of compounds 190-214 (one small error).

For Table S1, S2 and S3 (from Supplementary Materials) I have inserted page numbers.

There was a problem with references 138-140. I have corrected now the reference 138-Jain in reference 140, the reference 139-Urbonavicius in reference 138, and the reference 140-Bala in reference 139 in the subchapter references.

On page 5 in the text, I corrected 142-145 in 142-146, that is, I added the reference 146-Constantinescu.

Since I inserted the reference 146-Constantinescu in the text in Figure A and Figure B, the reference number changed from 146-Constantinescu to 66-Constantinescu.

We appreciate for Editors/Reviewers’ warm work seriously, and hope that the correction will meet with approval.

Yours sincerely,

Alin Mihis

Reviewer 4 Report

This review is a comprehensive survey of major scientific databases for information on the Anticancer Mechanisms of Chalcones based on a very detailed review of relevant scientific literature. The work is very interesting and well organized. I only have some minor observations that are listed in the following lines.

1.      There are some typos and grammatical errors throughout the manuscript. The work would benefit from close editing

2.      Line 11: Avoid using abbreviations in the abstract. Add what ABCG2 stands for at its first mention.

3.      The introduction is lengthy. Consider condensing the information in a more concise way.

4.      Methodology: Which databases were used? How were the references for this review selected? What were the reference inclusion and exclusion criteria?

5.      Summarize the compounds and their effects in a tabular form.

6.      Add a schematic diagram to summarize and compare the two mechanisms.

7.      The conclusion is too long and very vague. This section is meant to describe the main findings of the current study in a concise way. Please rewrite. 

Author Response

Hello!

Thank You very much for Your amiability and work concerning our manuscript, and we are grateful all reviewers’ comments on the manuscript. These comments are all very valuable and helpful in increasing the quality of our manuscript. We have studied these comments carefully and made correction in the revised manuscript according to all reviewers’ comments. These changes are performed with Track Changes (MS Word 2013) in the manuscript and in the Supplementary Materials. The detailed answers point by point are presented below:

Question 1:

There are some typos and grammatical errors throughout the manuscript. The work would benefit from close editing

Answer: Thank You for Your helpful suggestion. I have corrected several typos and grammar errors.

Question 2:

Line 11: Avoid using abbreviations in the abstract. Add what ABCG2 stands for at its first mention.

Answer: Thank You for Your helpful suggestion. I have rewritten the Abstract, I have modified ABCG2 in ATP binding cassette subfamily G.

Question 3:

The introduction is lengthy. Consider condensing the information in a more concise way.

Answer: Thank You for Your helpful suggestion.  I shortened the introduction by moving the data on the structure and biological activities of flavonoids to the Results chapter and the data on the structure and biological activities of chalcones to the Results chapter, thus creating 2 subchapters.

Question 4:

Methodology: Which databases were used? How were the references for this review selected? What were the reference inclusion and exclusion criteria?

Answer: Thank You for Your helpful suggestion. I have added Chapter 4, Methods:

The articles were selected from the PubMed and Google Scholar database, taking into account the most representative articles about cancer, flavonoids, about natural, semi-synthetic and synthetic chalcones with anticancer activity, with an emphasis on the mechanisms of ATP binding cassette subfamily G and tubulin. The inclusion criterion was that the articles were from the period 2010-2022 and the exclusion criterion was other mechanisms of anticancer action, different from ABCG2 and tubulin.

Question 5:

Summarize the compounds and their effects in a tabular form.

Answer: Thank You for Your helpful suggestion. I created a new table, S3, which contains half-maximal inhibitory concentration (IC50) of the most effective chalcones compounds with data from the publications.

Question 6:

Add a schematic diagram to summarize and compare the two mechanisms.

Answer: Thank You for Your helpful suggestion.

I have added a diagram for summarization: Figure 15. The two mechanisms and most active compounds (Page 17).

Question 7:

The conclusion is too long and very vague. This section is meant to describe the main findings of the current study in a concise way. Please rewrite.

Answer: Thank You for Your helpful suggestion. I have rewritten the conclusions.

At the recommendation of the expert reviewers, I made the following changes:

. I move Figure 1 and Figure 2 to the Supplementary Materials-Annex A, I have inserted in the text on page 2 and page 3 Figure A- Annex A (old Figure 1) and reference Constantinescu, T.; Lungu, C.N. Anticancer Activity of Natural and Synthetic Chalcones. Int. J. Mol. Sci. 2021, 22, 11306. I inserted in the text on page 20 Figure B- Annex A (old Figure 2) and reference Constantinescu, T.; Lungu, C.N. Anticancer Activity of Natural and Synthetic Chalcones. Int. J. Mol. Sci. 2021, 22, 11306.

I have added:

Figure 1. Structure of Hesperidin (page 3);

Figure 2. Structure of Xanthohumol (XN) (page 4);

Figure 3. Structure of Licochalcone A (page 6);

Figure 4. Structure of Licochalcone B (page 6);

Figure 5. General structure of chalcones 14-47 (page 7);

Figure 6. General structure of naphthochalcones (compounds 58-61) (page 8);

Figure 7. General structure of quinazoline chalcones (compounds 66-71) (page 8);

Figure 8. General structure of tariquidar-chalcones (compounds 132-137) (page 10);

Figure 9. Structure of a semisynthetic chalcone (compound 138) (page 12);

Figure 10. General structure of naphthalenchalcones (compounds 142-161) (page 13);

Figure 11. General structure of chalcones 162-177 (page 14);

Figure 12. General structure of chalcones 179-187 (page 15);

Figure 13. General structure of quinoline chalcones (compounds 190-214) (page 15);

Figure 14. General structure of shikonium-derived chalcones (compounds 215-232) (page 16);

Figure 15. The two mechanisms and most active compounds (page 17)

In addition, we made some changes in the manuscript using Track Changes option from MS Word 2013, which we list further:

I corrected the general structure of compounds 162-177 (there were 2 small errors) and the general structure of compounds 190-214 (one small error).

For Table S1, S2 and S3 (from Supplementary Materials) I have inserted page numbers.

There was a problem with references 138-140. I have corrected now the reference 138-Jain in reference 140, the reference 139-Urbonavicius in reference 138, and the reference 140-Bala in reference 139 in the subchapter references.

On page 5 in the text, I corrected 142-145 in 142-146, that is, I added the reference 146-Constantinescu.

Since I inserted the reference 146-Constantinescu in the text in Figure A and Figure B, the reference number changed from 146-Constantinescu to 66-Constantinescu.

We appreciate for Editors/Reviewers’ warm work seriously, and hope that the correction will meet with approval.

Yours sincerely,

Alin Mihis